# Combining 16S Sequencing and qPCR Quantification Reveals *Staphylococcus aureus* Driven Bacterial Overgrowth in the Skin of Severe Atopic Dermatitis Patients

**DOI:** 10.3390/biom13071030

**Published:** 2023-06-23

**Authors:** Amedeo De Tomassi, Anna Reiter, Matthias Reiger, Luise Rauer, Robin Rohayem, Claudia Traidl-Hoffmann, Avidan U. Neumann, Claudia Hülpüsch

**Affiliations:** 1Environmental Medicine, Faculty of Medicine, University of Augsburg, 86156 Augsburg, Germany; amedeo.detomassi@uni-a.de (A.D.T.); anna.reiter@tum.de (A.R.); matthias.reiger@uni-a.de (M.R.); luise.rauer@tum.de (L.R.); robin.rohayem@uni-a.de (R.R.); claudia.traidl-hoffmann@med.uni-augsburg.de (C.T.-H.); avidan.neumann@uni-a.de (A.U.N.); 2Institute of Environmental Medicine, Helmholtz Zentrum München, 86156 Augsburg, Germany; 3Environmental Medicine, Technical University of Munich, 86156 Augsburg, Germany; 4CK CARE, Christine-Kühne Center for Allergy Research and Education, 7265 Davos, Switzerland; info@ck-care.ch; 5ZIEL—Institute for Food and Health, Technical University of Munich, 85354 Freising, Germany

**Keywords:** absolute quantification, next-generation sequencing, qPCR, atopic dermatitis, *Staphylococcus aureus*, skin microbiome

## Abstract

Atopic dermatitis (AD) is an inflammatory skin disease with a microbiome dysbiosis towards a high relative abundance of *Staphylococcus aureus*. However, information is missing on the actual bacterial load on AD skin, which may affect the cell number driven release of pathogenic factors. Here, we combined the relative abundance results obtained by next-generation sequencing (NGS, 16S V1-V3) with bacterial quantification by targeted qPCR (total bacterial load = 16S, *S. aureus* = nuc gene). Skin swabs were sampled cross-sectionally (*n* = 135 AD patients; *n* = 20 healthy) and longitudinally (*n* = 6 AD patients; *n* = 6 healthy). NGS and qPCR yielded highly inter-correlated *S. aureus* relative abundances and *S. aureus* cell numbers. Additionally, intra-individual differences between body sides, skin status, and consecutive timepoints were also observed. Interestingly, a significantly higher total bacterial load, in addition to higher *S. aureus* relative abundance and cell numbers, was observed in AD patients in both lesional and non-lesional skin, as compared to healthy controls. Moreover, in the lesional skin of AD patients, higher *S. aureus* cell numbers significantly correlated with the higher total bacterial load. Furthermore, significantly more severe AD patients presented with higher *S. aureus* cell number and total bacterial load compared to patients with mild or moderate AD. Our results indicate that severe AD patients exhibit *S. aureus* driven increased bacterial skin colonization. Overall, bacterial quantification gives important insights in addition to microbiome composition by sequencing.

## 1. Introduction

Atopic eczema (also called atopic dermatitis, AD) is an inflammatory skin disease affecting up to 20% of children and 5% of adults [1,2]. The skin barrier is an interconnected network consisting of physical, chemical, immunological, and microbial aspects and is disrupted on multiple levels in AD patients [3]. Typically, a dysbiosis in the skin microbiome towards *Staphylococcus aureus* is observed and its relative abundance is associated with disease severity [4,5,6]. *S. aureus* encodes different virulence factors, such as toxins and proteases in its genome, which have a negative effect on the skin barrier. The expression of virulence factors such as toxins and biofilms are connected to AD severity and are regulated by quorum sensing and, hence, by the bacterial cell numbers [7,8,9,10,11].

While the analysis of the microbiome via 16S amplicon sequencing (next-generation sequencing, NGS) is a powerful method to analyze the microbial composition of a sample, the absolute bacterial load cannot be assessed with this standard procedure in scientific studies [12]. Even though the relative abundance may remain stable under certain conditions, the dynamics of absolute numbers can be very different [13] and have a biological implication due to the cell density-regulated expression of virulence factors via quorum sensing [14]. Thus, by neglecting absolute quantification data, underlying physiological or pathological mechanisms could remain hidden when only considering relative abundance [15].

To quantify the overall bacterial load or single taxa, additional cellular- and molecular-based methods such as fluorescence spectroscopy, flow cytometry, targeted quantitative PCR (qPCR), 16S qRT-PCR, ddPCR, and reference spike-in can be applied. While fluorescence spectroscopy, flow cytometry, and 16S qRT-PCR are suitable for the detection of viable cells, these methods are time-consuming in sample preparation and quantification [16]. As qPCR allows sensitive and specific enumeration of broad and specific microbial populations in a wide range of samples at the gene level in a rapid manner, it is a commonly used method in large scientific studies [16]. For the detection of the total bacterial load, the 16S rRNA is a typical target due to its conserved region across bacterial taxa. However, the 16S rRNA copy numbers vary strongly between different bacterial species [17,18,19]. Therefore, the copy number needs to be considered when calculating cell numbers. Alternatively, a unique gene for the identification of certain species can be chosen [20].

The aim of this study was to evaluate the consensus of *S. aureus* detection via qPCR and NGS. Furthermore, the relevance of bacterial load in the context of AD was assessed in a longitudinal and cross-sectional skin microbiome study comparing AD patients to healthy controls.

## 2. Materials and Methods

### 2.1. Microbiome Datasets

*LONGITUDINAL DATASET*. The longitudinal study investigated the skin microbiome of *n* = 6 healthy individuals (HE) and *n* = 6 AD patients weekly over the course of eight weeks while applying different emollients at the two body sides by using skin swabs. Patients were matched by age, gender, and sampling location. The BampH study was approved by the ethics committee of the Technical University of Munich (187/17S) [6]. Additionally, the raw microbiome data is publicly available at the European Nucleotide Archive under the study accession number PRJEB37663. In total, 108 AD samples were analyzed by qPCR and NGS, of which 1 AD sample did not pass NGS quality control. Furthermore, 90 HE samples were analyzed by qPCR after the exclusion of 1 individual in the HE due to an atopic condition. NGS of HE samples was not used here.

*CROSS-SECTIONAL DATASET*. The cross-sectional study includes samples from the ProRaD study which was approved by the ethics committee of Switzerland (EK2016-00301) and the local ethics committee of the Technical University of Munich 112/16S. ProRaD is a Prospective longitudinal study to investigate the Remission phase in patients with Atopic Dermatitis (AD) and other allergy-associated diseases, such as asthma, food allergies, and allergic rhinitis of the CK CARE consortium [21]. Here, *n* = 135 lesional skin swabs and *n* = 135 non-lesional skin swabs from the first study visit of AD patients from Augsburg (Ethics Technical University Augsburg 112/16S) were analyzed by NGS and qPCR. The lesional sample was taken from a region with the current lesion, preferably from the antecubital fossa (Ac). Two non-lesional samples did not pass NGS quality control.

*ADDITIONAL DATASET*. To increase the number of samples from healthy (HE) individuals, additional samples from *n* = 14 healthy individuals were included here from the UV study, which was approved by the ethics committee of the Technical University of Munich (57/18S, 112/16S). The UV study was set up to analyze the UV-protective effect of emollients on the skin physiology and microbiome in healthy participants. Here, only the non-UV radiated, non-treated, baseline visit samples from the volar arm (Va) of 14 HE participants were analyzed by qPCR.

*AD SEVERITY.* AD severity was determined via SCORingAtopicDermatitis (SCORAD), and patients were separated into three AD severity categories: mild (SCORAD from 0 to ≤25), moderate (SCORAD from >25 to ≤50), and severe (SCORAD >50) [22].

### 2.2. Microbiome Analysis

*SAMPLE PREPARATION.* For 16S rRNA gene amplicon sequencing, samples of all three skin microbiome datasets were prepared, as previously published [6]. In brief, skin swabs (Sigma-swab, MWE, Corsham, England) were taken and stored in 500 μL of Stool DNA Stabilizer solution (Stratec, Berlin, Germany). The DNA was extracted with the QIAamp UCP Pathogen kit (Qiagen: Hilden, Germany).

*16S rRNA GENE AMPLICON SEQUENCING.* The V1–V3 region was amplified using the 27F-YM (5-AGAGTTTGATYMTGGCTCAG-3) and 534R (5-ATTACCGCGGCTGCTGG-3) primers, and barcodes were added in a second PCR step. Subsequently, AMPure XP beads (Beckman Coulter, Fullerton, CA, USA) were used for amplicon purification. Samples were sequenced with the Illumina MiSeq^®®^ platform (Illumina Inc., San Diego, CA, USA) using 2 × 300 bp paired-end reads (MiSeq^®®^ Reagent Kit v3 600 cycles; Illumina Inc.). Denoising of the sequences was performed with DADA2 [23] and annotation with AnnotIEM [24] as well as with the standard RDP database formatted for DADA2 (https://doi.org/10.5281/zenodo.4310151 accessed on 25 April 2023). 

*QUANTIFICATION* VIA *QPCR.* Absolute quantification of the 16S rRNA gene and the unique *S. aureus* gene *nuc* were carried out via a TaqMan assay using the following primers and probes: *S. aureus* forward primer: GTTGCTTAGTGTTAACTTTAGTTGTA, reverse primer: AATGTCGCAGGTTCTTTATGTAATTT, and probe: FAM-AAGTCTAAGTAGCTCAGCAAATGCA-BHQ1 [25]; 16S rRNA gene copies forward primer: TGGAGCATGTGGTTTAATTCGA, reverse primer: TGCGGGACTTAACCCAACA, and probe: Cy5-CACGAGCTGACGACARCCATGCA-BHQ2 (Eurogentec S.A., Seraing, Belgium) [26]. The reactions were performed in 10 µL final volume using the PerfeCTa Multiplex qPCR ToughMix (Quantabio, Beverly, MA, USA) with a 100 nM concentration for each primer and probe in the multiplex setup. After a 2-min denaturation–activation step at 95 °C, 45 cycles were performed with a denaturation step of 15 s at 95 °C and an annealing–elongation step of 60 s at 60 °C in a CFX384 Real Time System (Bio-Rad Laboratories, Inc., Hercules, CA, USA). The quantity cycles (Cqs) were determined as the average of independent triplicates.

### 2.3. Statistical Analysis

The *S. aureus* cell number, also termed here as the qPCR-absolute-abundance, on the skin was assessed by determining the number of *nuc* gene copies, whereas the total bacterial load on the skin was assessed by determining the number of 16S gene copies, which notably is only a relative estimate since there is more than one 16S gene copy per bacterial cell. To compare the *S. aureus* relative abundance values obtained with NGS to qPCR results, the proportion of *S. aureus* cells (number of *nuc* gene copies) out of the total 16S gene copies per sample (*nuc* gene copies * 6/total 16S copies) was determined which we term as qPCR-relative-abundance, where the number of *nuc* gene copies is multiplied by 6 since the *nuc* gene appears only once per *S. aureus* cell, while *S. aureus* has, on average, 6 16S gene copies per cell [18].

Correlations between continuous variables were assessed using the Pearson test. Statistical significance of differences in continuous variable results between 2 and 3 groups was assessed with the non-parametric Mann–Whitney U Test or the Kruskal–Wallis test with Dunn’s pairwise post hoc test, respectively. Moreover, the statistical significance of differences in relative abundance between groups was assessed with the Fisher exact test for 2 × 2 contingency tables and with the Freeman–Halton extension for 2 × 3 contingency tables. *p*-values were considered significant at the alpha-error two-tailed level of *p* < 0.05. No correction for multiple testing was applied since we had only tested 3 pre-defined variables in this analysis (*S. aureus* relative abundance, *S. aureus* absolute abundance, and 16S copy number). However, even if the strict Bonferroni correction had been applied for the 3 variables, all the relevant *p*-values would still have been highly significant as they are mostly *p* < 0.0001. Unless otherwise stated, median values are shown in the figures. The statistical analyses and plots were performed and created, respectively, either with R Version R-4.2.1 or GraphPad Prism version 9.5.0 for Windows (GraphPad Software, La Jolla CA, USA).

## 3. Results

### 3.1. Consensus in S. aureus Abundance Results Via NGS and qPCR

To assess the consensus of the results obtained via NGS and qPCR, the relative and/or absolute abundance of *S. aureus* were determined with both methods in lesional skin samples from AD patients in both the longitudinal skin microbiome dataset (*n* = 6) and the cross-sectional dataset (*n* = 135) (demographic data is included in Appendix A).

The absolute *S. aureus* abundance measured by qPCR of the *nuc* gene correlated strongly and significantly with the relative abundance of *S. aureus* observed with NGS for lesional AD samples with detectable *S. aureus* in the longitudinal and cross-sectional datasets (Figure 1). Similarly, the detected *S. aureus* relative abundance of AD lesional samples was highly and significantly correlated by qPCR and NGS in both datasets (Appendix A). However, in both datasets, a proportion of the samples had a discrepancy in *S. aureus* detection between the two methods, with *S. aureus* detected either only by qPCR or by NGS. This phenomenon was especially observed in the low range of relative *S. aureus* abundance <1%. While the longitudinal dataset had a higher number of samples with undetectable *S. aureus* by NGS, the cross-sectional dataset had a higher number of samples with undetectable *S. aureus* via qPCR (Figure 1). The sensitivity of qPCR relative to NGS was 97.8% in the longitudinal dataset and 72.1% in the cross-sectional dataset, while the corresponding sensitivity of NGS relative to qPCR was 88.1% in the longitudinal dataset (Figure 1A) and 100% in the cross-sectional dataset (Figure 1B).

Moreover, the intra-individual differences in *S. aureus* relative abundance were strongly reproducible between the NGS and qPCR methods when comparing the difference between body sides in the longitudinal dataset (Figure 2A) or between lesional and non-lesional samples in the cross-sectional dataset (Figure 2B).

The changes in the longitudinal behavior of *S. aureus* follow the same pattern in relative abundance as measured by NGS, relative abundance as estimated by qPCR, or absolute abundance measured by qPCR (Figure 3A,B). However, the amplitude of fluctuations is higher when measured via qPCR than via NGS, as seen in the example in Figure 3A, which was validated for all patients (Figure 3B, Appendix A).

### 3.2. Bacterial Load in Atopic Dermatitis

The total bacterial load in healthy individuals and in atopic dermatitis patients was estimated by the number of 16S rRNA gene copies measured by qPCR. The longitudinal dataset with six AD patients and five healthy individuals revealed highly similar 16S rRNA gene copy numbers between body sides per individual in the HE and AD groups (Figure 4A,B). The individual bacterial load remained relatively stable throughout the study period of eight weeks. Interestingly, a higher variation in bacterial load was observed between the healthy individuals as compared to AD patients, but nevertheless, all AD patients showed higher bacterial load than HE controls (Figure 4A,B).

Combining data from the longitudinal, cross-sectional, and additional healthy datasets, the total bacterial load was significantly higher in AD non-lesional (*n* = 135) and lesional (*n* = 135) skin compared to healthy (*n* = 20) individuals (HE median = 8.3 × 10^4^, AD NL median = 3.5 × 10^5^, AD LS median = 5.5 × 10^5^) (Figure 4C).

To understand whether the higher bacterial load measured in AD patients is due to an overgrowth of *S. aureus* or a general increase in bacterial load, the number of measured 16S rRNA gene copies was correlated with the number of *S. aureus* cells. Strikingly, a significant correlation between the total bacterial load (16S copy number) and *S. aureus* absolute abundance (number of *nuc* gene copies) was observed only in the lesional samples of AD patients with *S. aureus* absolute abundance larger than 10^5^ but not for patients with *S. aureus* absolute abundance lower than 10^5^ (Figure 4D). This observation indicates that a high bacterial load in AD skin is driven by an increasing number of *S. aureus* cells.

### 3.3. Association between S. aureus Bacterial Colonization and Atopic Dermatitis Severity

To investigate whether AD disease severity is driven by the proportion of *S. aureus* within the bacterial community or by absolute *S. aureus* cell numbers, we analyzed *S. aureus* abundance and the total bacterial load in the lesional samples from the cross-sectional dataset (*n* = 135) based on the following AD severity categories: mild, moderate, and severe, as defined by the SCORAD index. Higher *S. aureus* relative abundance, measured by NGS, as well as *S. aureus* absolute abundance measured by qPCR, showed a significant association with AD severity (Figure 5A,B). In addition, significantly more mild AD patients (68.9%) had undetectable *S. aureus* load by qPCR as compared to patients with moderate (33.3%) and severe (15.4%) AD (Figure 5D). The fraction of patients with undetectable *S. aureus* load was not significantly different between moderate and severe patients.

Interestingly, while the median *S. aureus* absolute abundance in mild AD patients was undetectable via qPCR, the median was slightly higher in moderate AD (70 cells) and significantly higher in severe AD (median 3 × 10^5^ cells) patients (Figure 5B). Significantly more severe AD patients (61.5%) also had absolute *S. aureus* abundance larger than 10^5^, the threshold identified for correlation with total bacterial load (Figure 4D), as compared to patients with moderate (17.6%) and mild (11.1%) AD (Figure 5D).

Moreover, the total bacterial load (16S copy number) was significantly higher in severe AD patients (median 2 × 10^6^) as compared to moderate AD (5 × 10^5^) and mild AD (6 × 10^5^) (Figure 5C). A total bacterial load larger than 10^6^ was observed in significantly more (66.7%) severe AD patients, as compared to patients with moderate (35.3%) and mild (37.8%) AD (Figure 5D). Thus, severe AD patients demonstrate a higher *S. aureus* abundance associated with a higher total bacterial load.

*S. aureus* absolute and relative abundance as well as 16S copy numbers were not significantly associated with gender and not correlated with age (Appendix A). Furthermore, the association of severe AD with higher levels of *S. aureus* absolute and relative abundance, as well as with higher 16S copy numbers, is observed independently of the age range (Appendix A).

## 4. Discussion

Our results suggest a high level of consistency in the relative abundance of *S. aureus* between data generated via qPCR and NGS. Nonetheless, the absolute quantification of total bacterial load and of *S. aureus* gave additional relevant biological insights, hinting towards an *S. aureus* driven total bacterial overgrowth in absolute cell numbers in severe AD patients.

Even though there was a strong correlation between the *S. aureus* relative and absolute abundances determined via NGS and qPCR, some samples had *S. aureus* detected either only by qPCR or only by NGS. Both methods used in this study were DNA-based approaches and therefore are comparable techniques. While DNA-based methods such as NGS and qPCR may be affected by biases introduced during sampling, cell lysis and DNA extraction, these steps would not explain the differences between the two detection methods since both use the same extracted bacterial DNA. Therefore, the integrity of the DNA is important for reliable results in both methods [27].

There is a wide range of possible explanations for the discrepancy of observed *S. aureus* relative and absolute abundances between the two methods. First, both use different amounts of input DNA, where stochastic effects can particularly vary the detection of low-range *S. aureus* abundance. Discrepancies were mainly observed in low-level *S. aureus* abundance and therefore close to the technical limit of detection and quantification [28]. Furthermore, two different target genes were used for qPCR detection of *S. aureus* (*nuc*, single copied) and NGS detection of *S. aureus* (16S rRNA gene copies, median of six copies). The difference in copy numbers could explain the failed qPCR detection of *S. aureus*, while NGS gave a positive result for *S. aureus* based on the multi-copy 16S rRNA gene [18,19]. Additionally, mutations in one of the target genes per method could lead to reduced amplification in PCR for NGS or qPCR, and would affect only one of the two methods. Moreover, a higher calculated percentage of *S. aureus* in relation to 16S rRNA gene copies was found in some samples, possibly due to *S. aureus* strains with either more than six copy numbers of 16S rRNA gene copies or reduced efficiency of the 16S rRNA gene qPCR reaction.

A particular challenge for NGS data is the taxonomic (mis)annotation of sequences. In this study, the V1–V3 region of the 16S rRNA was sequenced and used for species annotation. Using only parts of the 16S rRNA gene for species annotation is not as precise as using the full-length 16S rRNA gene [29]. However, the V1–3 hypervariable region represents bacterial communities similarly well as the whole genome sequencing [30], and is able to distinguish *S. aureus* from most other Staphylococci [29,30]. To achieve a highly accurate species-level classification, we used taxonomic assignment by AnnotIEM, which combines several databases for annotation [24,31]. To verify the annotation given by AnnotIEM based on multiple databases, each sequence was re-annotated using the standard RDP database through DADA2, which confirmed the *S. aureus* annotation.

Even though qPCR is frequently used in science due to its aforementioned advantages, qPCR also suffers from challenges, such as reduced sensitivity due to PCR inhibitors or the multiplex set-up. Particularly when close to the detection limit, the precision of qPCR might be hampered as low-level presence of the target gene can result in stochastic amplification and measurement uncertainties [32]. A coefficient of variation between 28% and 79% was observed for three test organisms near the detection limit due to instrument repeatability in one study [33]. Another study revealed that a relative abundance of <10% was associated with high rates of errors when inferring the original species concentration [34].

Regarding the biological implications of total bacterial cells on the skin, the longitudinal dataset revealed a participant-specific bacterial load in healthy individuals between 10^3^ and 10^6^ copy numbers of 16S rRNA. A bacterial load between 10^3^ and 10^7^ 16S rRNA gene copies has been reported on healthy skin, with major site-specific differences between moist, sebaceous, and dry body sides [35]. Additionally, in the longitudinal dataset, different body locations were sampled, which might have affected the large variation in bacterial colonization observed in healthy individuals. However, these differences between either individuals or body sides were lost in the AD patients, where all patients had a median of 10^5^ bacterial cells on the sampled skin location.

The total bacterial load in AD patients was increased 10-fold compared to healthy individuals in our data as well as in previous studies [35,36]. In other environments, such as the gut, the importance of the overall bacterial load in human physiology and health has already been reported [13,37,38]. Differences in bacterial load are plausibly associated with disease status, as known from single-pathogen infectious diseases and the effect of quorum sensing on the pathogenicity of bacteria. In the context of AD, an increase in *S. aureus* relative abundance, especially during disease flares in lesional skin, is already well known and could be confirmed in this study [4]. *S. aureus* virulence factors, such as toxins, proteases, and biofilms, are known to negatively impact the skin barrier and exacerbate AD severity [39]. The release of toxins and proteases by *S. aureus* highly depends on quorum sensing, and therefore the absolute bacterial cell numbers might hypothetically be more relevant than the relative composition of microbes on the skin [7,14]. However, other bacterial species, fungal infections, IgA and IgE levels, and other clinical factors (such as other allergy-associated diseases, asthma, food allergies and/or allergic rhinitis) may also be associated with AD severity but were not further analyzed here.

In this study, we show, for the first time, that in AD patients, not only the bacterial composition is changed towards *S. aureus* but also that there is an *S. aureus* driven increase of the total bacterial load, especially in severe patients. Strikingly, the bacterial load increased in an *S. aureus* driven manner only above a threshold of 10^5^
*S. aureus* cells. This could be a critical cell number to activate the expression of toxins and proteases, leading to inflammation and skin barrier disruption, which can lead to an increased skin pH [40]. Changes in the microenvironment such as an increased skin pH are in favor of *S. aureus*, leading to an increased growth rate, followed by disease exacerbation due to further irritation of the skin by virulence factors [6,9,40]. In line, successful AD treatment with the monoclonal, IL4-Rα- blocking antibody Dupilumab has been reported to reduce absolute bacterial load and *S. aureus* load and to increase microbial diversity while reducing disease severity [41,42].

The evidence presented here for *S. aureus* driven bacterial overgrowth in AD provides mechanistic insights into the development and exacerbation of AD symptoms and may indicate *S. aureus* and its quorum sensing system as a target for new therapeutics in AD. We demonstrated in this study that NGS data on the skin microbiome composition can be importantly complemented by information on absolute cell number provided by quantitative methods, such as qPCR in the context of AD research.

## Figures and Tables

**Figure 1 biomolecules-13-01030-f001:**
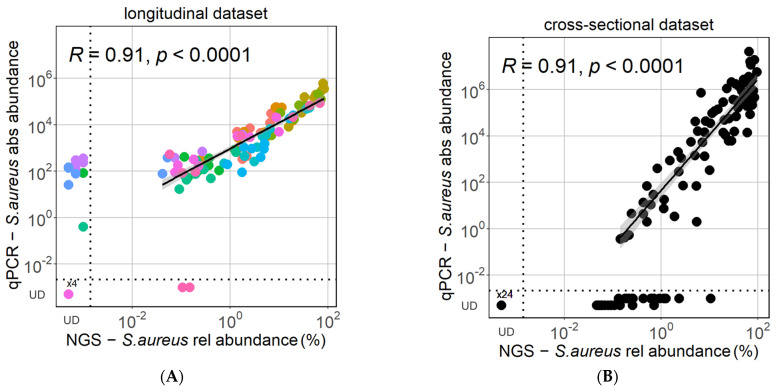
High consensus of qPCR and NGS in estimating *S. aureus* abundance. There is a strong correlation between the relative *S. aureus* abundance observed via NGS and the absolute *S. aureus* abundance measured by qPCR in both the longitudinal (**A**) and the cross-sectional (**B**) datasets. Pearson correlations were performed for samples with detectable *S. aureus*. Colors in the longitudinal dataset correspond to samples from the 6 different patients. UD = undetectable.

**Figure 2 biomolecules-13-01030-f002:**
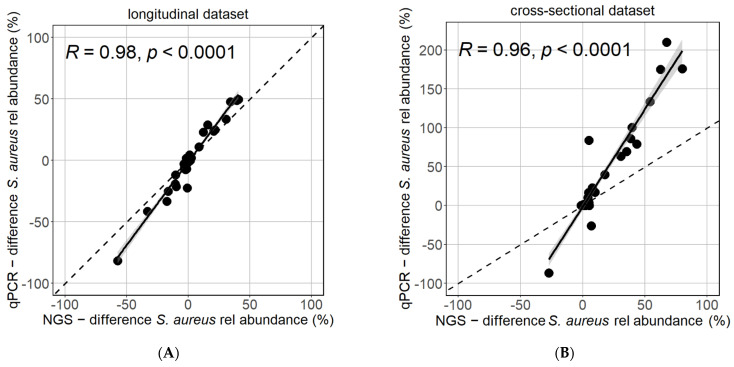
Reflection of intra-individual differences is similar between NGS and qPCR. The difference in *S. aureus* relative abundance between body sides (**A**) or between lesional and non-lesional samples (**B**) of same patients at the same timepoints was strongly and highly significantly correlated (Pearson test).

**Figure 3 biomolecules-13-01030-f003:**
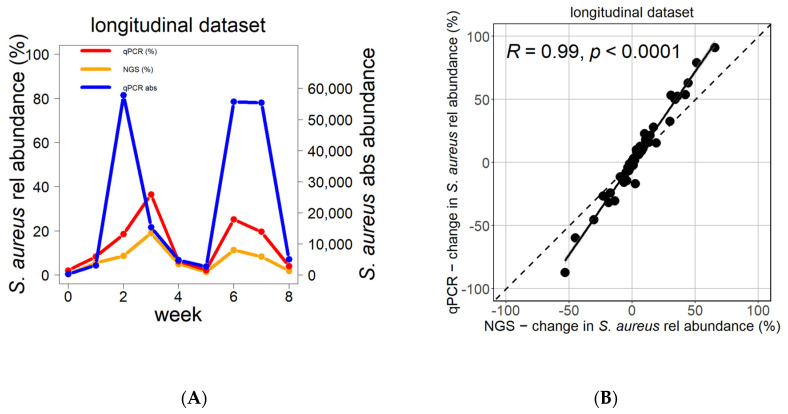
Change in *S. aureus* abundance over time as reflected by NGS and qPCR. The changes in relative and absolute *S. aureus* abundance over time followed the same pattern as measured by NGS or by qPCR (**A**). The differences in *S. aureus* relative abundance between consecutive time points significantly correlated between NGS and qPCR (**B**). Blue reflects absolute *S. aureus* abundance (qPCR), red reflects relative abundance of *S. aureus* by qPCR, and orange indicates relative abundance of *S. aureus* by NGS. Correlation analysis was performed using the Pearson test.

**Figure 4 biomolecules-13-01030-f004:**
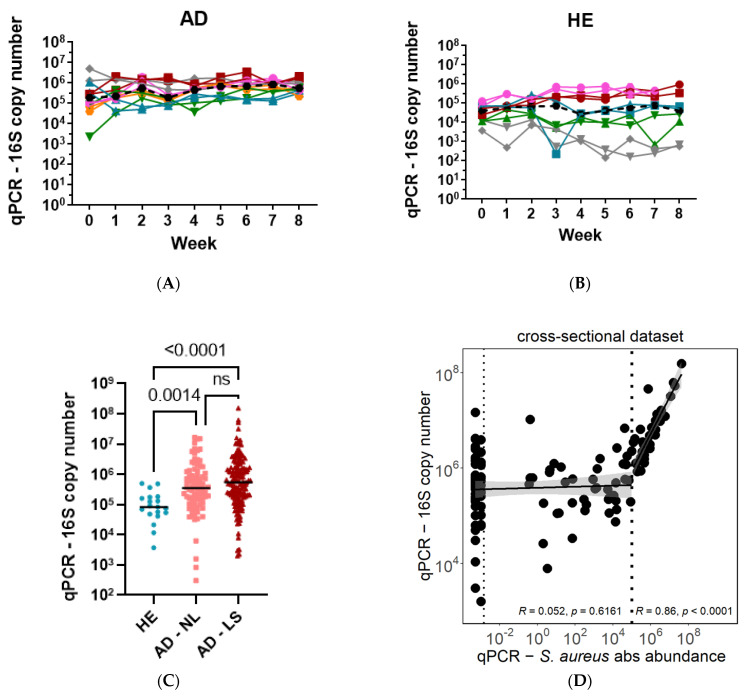
Bacterial overgrowth driven by *S. aureus* in AD patients. The bacterial load measured as 16S rRNA gene copies remained relatively stable in AD patients (**A**) and healthy controls (**B**) over nine weeks on two body sides. Overall, bacterial load is significantly higher in AD non-lesional (NL) and lesional (LS) skin compared to healthy (HE) controls (**C**). Total bacterial load (16S copy number) is correlated with *S. aureus* absolute abundance in lesional skin of AD patients only for patients with *S. aureus* absolute abundance larger than 10^5^ (**D**). In (**A**,**B**), colors represent different patients and shapes show body sides. ns = non-significant, UD = undetectable.

**Figure 5 biomolecules-13-01030-f005:**
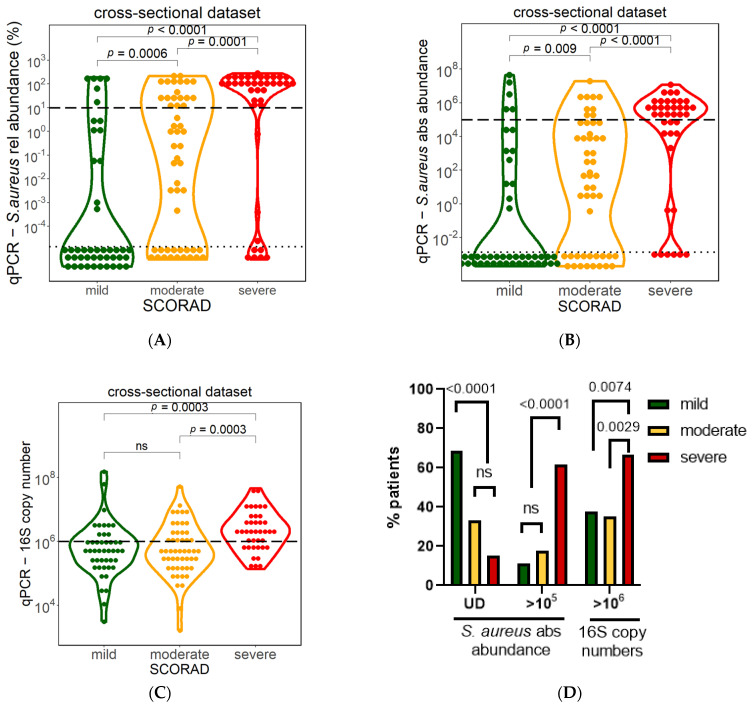
Association between *S. aureus* and AD severity. *S. aureus* relative abundance measured via NGS (**A**) and *S. aureus* absolute abundance measured via qPCR (**B**)) were significantly higher in severe AD patients. Bacterial load of severe patients determined via 16S rRNA gene copies was significantly higher in severe patients compared to mild and moderate patients (**C**). Significant differences were observed in numbers of patients with undetectable or high *S. aureus* and total bacterial load between mild, moderate, and severe patients (**D**). Mild= SCORAD between 0 and 25, moderate= SCORAD between >25 and 50, severe= SCORAD >50. Mann–Whitney U test was used for statistical analysis. Dotted horizontal lines indicate the threshold for detection limit (UD = undetectable), and dashed horizontal lines denote high numbers of *S. aureus* (10^5^ for absolute *S. aureus* abundance and 10% for relative *S. aureus* abundance in qPCR or NGS). ns = not-significant, UD = undetectable.

## Data Availability

Data has been partly published already and is accessible in the European Nucleotide Archive under the accession number PRJEB37663.

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
