# Peer review of "Combining 16S Sequencing and qPCR Quantification Reveals Staphylococcus aureus Driven Bacterial Overgrowth in the Skin of Severe Atopic Dermatitis Patients"

_biomolecules, 2023, doi:10.3390/biom13071030_

Round 1

Reviewer 1 Report

Line 180: what's the meaning of "Dunns correction for multiple testing?" Do the authors mean they applied Dunn's pairwise post hoc test? If yes, which correction method for multiple comparisons has been applied to p values?

Lines 258-259:  The sentence is not clear, authors should better explain their conclusion on the basis of observed results.

Figure 4: please add the meaning of UD in the figure caption

Line 272: please remove "(mild: SCORAD 0 to ≤25, moderate >25 to ≤50 and severe >50)" since this information has already been reported in the material and method section

Line 279-282: please rephrase the sentence, since it is not clear at all.

Line 290-293: please rephrase the sentence since it is not clear at all

Figures: while reporting p values, authors should avoid reporting sometimes exact values and sometimes thresholds. I suggest modifying images by introducing notations like a series of asterisks to denote levels of statistical significance (e.g. *: p<0.05; **: p<0.001, ***:p<0.0001 etc…)

Please remove the p values thresholds reported along the whole text (e.g line 277) to indicate statistical significance. If absolutely needed, in those contexts Authors should report exact values!

Author Response

Dear Reviewer,

thank you for your valuable feedback. Please see attachment for our Point-to Point-Response.

Reviewer 2 Report

The authors Tomassi et al., have studied the bacterial abundance in Atopic dermatitis (AD) skin. Using next-generation sequencing (NGS) and with targeted qPCR they have quantified and confirmed the bacteria load. Also, the authors were able to study the longitudinal and cross-sectional skin microbiome study comparing AD patients to healthy controls.

Minor Revisions:

Inconsistency in figure numbering. Can you please include the A, B, C numbering in Figure 2, 3, 4, and 5 like figure 1. It will increase the readability

Author Response

(The authors gave the same response as above.)

Reviewer 3 Report

biomolecules-2392848 

Combining 16S sequencing and qPCR quantification reveals Staphylococcus aureus driven bacterial overgrowth in the skin of severe atopic dermatitis patients.

This is an interesting paper trying to evaluate the consensus of Staphylococcus aureus detection via qPCR and next-generation sequencing.

But, several aspects should be addressed:

1.- The authors only include six patients (over the course of eight weeks).

2.- The authors do not specify the age of every patient (and healthy) individuals, as well as the possible influence of Streptococcus during this study.

3.- The comorbidity is not indicated in everyone (such as other allergy-associated diseases, asthma, food allergies and/or allergic rhinitis).

4.- Some correlation with IgA deficit should be considered, including the participation of fungal pathologies.

5.- Conclusions remarks must be emphasized.

Author Response

(The authors gave the same response as above.)

Round 2

Reviewer 3 Report

The authors have follow the suggested changes.